# The Feasibility of Sentinel Lymph-Node, Mapped with Indocyanine Green, Biopsy in Endometrial Cancer Patients: A Prospective Study

**DOI:** 10.3390/medicina58060712

**Published:** 2022-05-26

**Authors:** Migle Gedgaudaite, Arturas Sukovas, Saulius Paskauskas, Arnoldas Bartusevicius, Vaida Atstupenaite, Eimantas Svedas, Joana Celiesiute, Arvydas Cizauskas, Daiva Vaitkiene, Adrius Gaurilcikas

**Affiliations:** 1Department of Obstetrics and Gynaecology, Medical Academy, Lithuanian University of Health Sciences, LT-44307 Kaunas, Lithuania; arturas.sukovas@lsmuni.lt (A.S.); saulius.paskauskas@lsmuni.lt (S.P.); arnoldas.bartusevicius@lsmuni.lt (A.B.); eimantas.svedas@lsmu.lt (E.S.); joana.celiesiute@lsmu.lt (J.C.); daiva.vaitkiene@lsmuni.lt (D.V.); adrius.gaurilcikas@lsmu.lt (A.G.); 2Department of Radiology, Medical Academy, Lithuanian University of Health Sciences, LT-44307 Kaunas, Lithuania; vaida.atstupenaite@lsmu.lt; 3Department of Pathological Anatomy, Medical Academy, Lithuanian University of Health Sciences, LT-44307 Kaunas, Lithuania; arvydas.cizauskas@lsmu.lt

**Keywords:** endometrial cancer, sentinel lymph-node biopsy, indocyanine green, sentinel lymph-node mapping

## Abstract

*Background and objectives.* Systematic pelvic lymphadenectomy (LND) is an essential part of lymph-node status evaluation in endometrial cancer (EC) patients to tailor the adjuvant treatment. However, it is associated with the post-operative lymphatic complications and does not improve the outcomes of the disease. Indocyanine green (ICG) mapped sentinel lymph-node biopsy (SLB) has recently been introduced into the clinical practice as an alternative for the surgical lymph-node evaluation in EC patients with the potential to decrease LND related complications. The aim of our study was to evaluate the feasibility of ICG mapped SLB in low, intermediate, and high-risk EC patients in a center with no previous experience on endoscopic SLB procedure. *Materials and Methods*: The prospective study was performed. 170 patients with histologically confirmed EC were included. Sentinel lymph-nodes (SLs) were mapped with ICG dye and removed ahead of the total laparoscopic hysterectomy. Low-risk patients received only SLB, while SLB and LND were performed for intermediate and high-risk patients. *Results*: The overall detection rate of SLs was 88.8%. Bilateral mapping was achieved in 68.2% of the patients. The overall detection rate for low-risk patients was 93.7%, 85.0% for the intermediate-risk group, and 100% for high-risk patients (*p* = 0.232). The most common anatomical sites of SLs were the external iliac (45.8% on the right and 46.6% on the left) and obturator regions (20.9% and 25.6%, respectively). Positive lymph-nodes were found in 8 (4.7%) patients. The sensitivity of SLB was 75.0% and negative predictive value (NPV)—97.2%. *Conclusions*: Even in the center with no previous experience, sentinel lymph-node biopsy using ICG mapping is feasible. However, the favorable outcomes might be associated with the learning process of newly established method.

## 1. Introduction

The prevalence of lymph-node metastasis in endometrial cancer (EC) patients varies from 10% in endometrioid-type cancer to 40% in non-endometrioid type of uterine tumours. The main risk factors for lymph-node involvement are as follows: high tumour grade, deep myometrial invasion, and tumour size larger than 2 cm [1]. Even though systematic pelvic lymphadenectomy (LND) does not improve the outcomes of the disease, surgical evaluation of the lymph-nodes is an essential part in assessing the need for adjuvant treatment in EC patients [2]. However, LND remains one of the most important factors associated with post-operative lymphatic complications that negatively impact the patients’ quality of life [3,4].

Recently, the indocyanine green (ICG) mapped sentinel lymph-node biopsy (SLB) in EC has been introduced into clinical practice. The results of clinical trials are promising [5,6,7,8,9,10,11] and a potential to decrease lymphatic complication rates has been demonstrated [12].

The SLB concept is based on the uterine lymphatic anatomy and its success depends on the ability to implement proposed protocols [13]. However, currently there is no unanimously acknowledged protocol for this procedure [2,14].

The aim of our study was to evaluate the feasibility of ICG mapped SLB in low, intermediate, and high-risk EC patients in a center with no previous experience of this procedure.

## 2. Materials and Methods

### 2.1. Design

This is a prospective interventional study, performed in the Lithuanian University of Health Sciences Hospital, Kaunas Clinics, Department of Obstetrics and Gynaecology during the period of March 2018 and December 2021.

### 2.2. Patients

Patients with histologically confirmed primary endometrial carcinoma and planned surgical treatment were chosen to participate in the study.

The standard treatment protocol included total hysterectomy with bilateral salpingo-oophorectomy and surgical lymph-node evaluation. Depending on the pre-operative risk, either sentinel SLB alone or SLB together with LND were performed.

### 2.3. Pre-Operative Risk Assessment and Lymph-Node Evaluation Type Selection

Pre-operative risk was assessed according to ESGO/ESTRO/ESP Endometrial Cancer guidelines [2]. To determine the pre-operative risk, preliminary histological evaluation (tumour histological form with differentiation grade (G)) and ultrasound exam results (the implied myometrial and cervical invasion) were used:Low risk: G1 or G2 with minimal myometrial invasion (less than half of the myometrium).Intermediate risk: G1 or G2 with myometrial invasion of more than half of the myometrium; or G3 with less than half of myometrial invasion.High risk: G3 with myometrial invasion of more than half of the myometrium.

According to the pre-operative risk, the type of surgical lymph-node evaluation was chosen:For low-risk patients—only the SLB procedure.For intermediate and high-risk patients—SLB followed by LND.

The flow chart of the patients’ selection is presented in Figure 1.

We collected data concerning the patients’ age, weight, body mass index (BMI), extragenital pathology, pre-operative ultrasound exam, and tumour histological assessment. To evaluate surgical morbidity, data about surgery time, blood loss, and intraoperative complications, the rate of conversion to laparotomy was gathered. The data about sentinel lymph-nodes sites and post-operative histological evaluation was used to assess the SLB procedure. Postoperative complications were classified according to the Clavien-Dindo classification [15].

### 2.4. Sentinel Lymph-Node Mapping Technique

The sentinel lymph-node (SL) mapping with ICG dye was performed using a technique previously described by Geppert B. et al., [13] and approved by our department. We used VERDYE^®^ (Green Diagnostic GmbH, Aschheim, Germany) 2.5 mg/mL powder to produce an injectable solution. A total of 25 mg of active substance was diluted with 10 mL of sterile injection water.

In the operating theatre, under general anaesthesia and after vaginal preparation, 1 mL of prepared dye was injected into 4 quadrants (2–4–8–10 o’clock) of the uterine cervix (0.25 mL each). Half of the dye was injected submucosally and the other half—1–3 cm into the stroma. Next, a uterine manipulator (RUMI, Cooper Surgical, Trumbull, CT, USA) was introduced.

During endoscopic surgery ICG mapped SLs were visualized with an OLYMPUS^®^ VISERA Elite II CLV-S200-IR system (OLYMPUS Corporation, Tokyo, Japan), using a pre-installed near-infrared light camera. After opening the avascular retroperitoneal spaces, near-infrared camera mode was activated, full inspection of iliac region was performed, visualizing obturator, external iliac, internal iliac, and common iliac regions for mapped sentinel lymph-nodes with afferent and efferent lymphatic vessels. The mapped SLs were removed separately, while documenting the anatomical site.

The lymph-nodes that were not mapped with ICG but were suspected to be pathological by macroscopic assessment were also removed separately.

After the SL removal, surgery continued. For intermediate and high-risk patients’ systemic lymphadenectomy of the pelvis was performed removing lymph-nodes from obturator region, external, internal, and common iliac regions. If there was a suspected pathological lymph-nodes in the para-aortic region up to inferior mesenteric artery, they were removed separately as well.

All surgeries were performed by eight different surgeons with no previous experience on SLB procedure with ICG in laparoscopic surgery.

### 2.5. Statistical Analysis

All calculations were made using Statistical Package of Social Sciences, Mac version 27 (SPSS, IBM, Brøndby, Denmark). Continuous variables without normal distribution were described using median and interval, while categorical variables, such as demographic and clinical characteristics, were reported as frequency and percentage. A chi-square test was used to compare the categorical variables.

The sensitivity, false negative rate (FNR), and negative predictive value (NPV) were calculated by results of the patients in the SLB + LND group, as each of them could be their own self-control group. The test was considered as a true positive if at least one mapped SL had a metastasis found during histological evaluation, and as a false negative, if there was no metastatic disease in SL, but metastasis was found in other lymph-nodes. The sensitivity was calculated as the proportion of true positive lymph-nodes and total number of patients with metastatic nodes; FNR—as proportion of false negative lymph-nodes among all patients with metastatic nodes; and NPV—proportion of true negative lymph-nodes among total number of patients with performed SLB. Specificity was not calculated, since there could be no false positive results in this type of study.

The statistical significance level was *p* value of less than 0.05.

## 3. Results

### 3.1. Patient Characteristics and Surgery-Related Data

A total of 170 patients were included in the study. General characteristics of the study population are presented in Table 1.

All patients except one had histologically confirmed endometrioid-type tumours, having been confirmed either after uterine curettage (76.5%), pipelle biopsy (21.2%), or after endometrial polypectomy (2.4%). Only one patient was diagnosed with serous adenocarcinoma and was assigned to a high-risk group.

All the patients with intermediate and high pre-operative risk (*n* = 107) were assigned to a SLB and full LND group. However, due to technical or medical concerns during surgery, full LND was abandoned in 17 of them and only SLB was performed. They were re-assigned to a SLB only group for result analysis.

The parameters of surgical performance are presented in Table 2.

Both groups had a case of conversion to laparotomy after the identification and removal of sentinel lymph-nodes, therefore both patients were included into the study. In one case (SLB + LND group), carcinosis of the peritoneum was suspected and laparotomy was performed to achieve optimal cytoreduction; however, severe endometriosis of the peritoneum was confirmed in the final pathology report. In the other case (SLB only group), laparotomy was performed to extract a big-volume uterus in order to avoid the spillage of content into the vagina.

Only seven (4.1%) patients had post-operative complications: one patient had vaginal lymphorrhea, three patients had post-surgical infection requiring antibiotic therapy, and two patients in the SLB + LND group had post-surgical peritonitis and sepsis due to bowel injuries and required re-laparotomies. In the SLB-only group one patient had an intraoperative bowel injury that happened while extracting the uterus. The lesion was identified immediately and sutured laparoscopically. The post-operative period was un-eventful.

### 3.2. Sentinel Lymph-Node Detection Rate and Anatomical Sites

Successful bilateral SL detection was achieved in 68.2% (116/170) of the patients. Additionally, 20.6% (35/170) of the patients had their SLs mapped unilaterally. Overall detection rate of SLs was 88.8%.

The median time from ICG injection until the dissection of the first SL was 30 min (interval 13–60 min), while 45 min (interval 20–90 min) for the contralateral one.

The SL detection rate, according to pre-operative risk, is presented in Figure 2. The overall detection rate for low-risk patients was 93.7%, 85.0% for the intermediate-risk group, and 100% for high-risk patients. There was no statistical difference between the groups (*p* = 0.232).

SL detection rates, according to the study period, are presented in Figure 3. Overall and bilateral detection rates improved over the period of the study, despite the difference being insignificant (*p* = 0.593).

The anatomical sites of SLs are shown in Figure 4. The most common anatomical sites of SLs in the right hemipelvis were external iliac region (45.8%), obturator region (20.9%), and internal iliac region (19.0%). In the left hemipelvis the locations of the SLs were similar —external iliac region (46.6%), obturator region (25.6%), and internal iliac region (17.3%). In three (1.8%) cases the SL was found in the paraaortic region.

The anatomical sites of mapped SLs in endometrioid and non-endometrioid type of tumours are presented in Table 3. In the endometrioid type of tumours most common SL sites were external iliac (45.2%) and obturator regions (21.8%), while in non-endometrioid tumours SLs were most frequently detected in obturator region (50.0%); however, this difference was not significant (*p* = 0.352).

### 3.3. Lymph-Node Histological Evaluation: Sensitivity and Negative Predictive Value of SLB

Full LND was performed in 90 (52.9%) patients. All of them were either in the intermediate, or high pre-operative risk group. The median number of removed lymph-nodes was 7.5 (interval 3–22).

Positive lymph-nodes were found in eight (4.7%) patients.

The sensitivity and negative predictive value of the SLB procedure was calculated using the results of 77 patients; of that, 76 of them had at least one mapped SL and systematic LND. As there cannot be false-positive results in this type of study, we also included one patient, who, due to being assigned to the low-risk group pre-operatively, did not undergo full LND but had a positive SL.

The sensitivity of SLB was 75.0% and NPV–97.2% (Table 4).

**Table 4 medicina-58-00712-t004:** Accuracy of SL biopsy.

	True Positive Node	True Negative Nodes
Positive SLN	6	0
Negative SLN	2	69

The trend of FNR over the study period is presented in Table 5.

### 3.4. Final Histological Tumour Evaluation

Final histological tumour evaluation data and post-operative risk assessment are presented in Table 6.

The comparison of risk assessment before and after the surgery is presented in Table 7.

## 4. Discussion

This study was conducted to evaluate the feasibility of sentinel lymph-node biopsy in endometrial cancer patients in an Oncogynaecological Centre with no previous experience on SLB in laparoscopic surgery. The surgeries were completed by eight different surgeons who started to perform SLB procedures at the same time. We believe this trial could be considered as “the real-life study” on implementation of the new method in the clinical practice.

The SLB is a well-established procedure in some cancers, such as breast, vulvar cancer, or melanoma. It allows to tailor the adjuvant treatment, while evading complications associated with systemic lymphadenectomy. In cervical cancer, a number of studies have demonstrated the feasibility and accuracy of SLB. When ICG based minimally invasive technique of SLB was developed, SL evaluation became increasingly utilized as part of surgical staging in endometrial cancer [16].

The result of our study showed the overall SL detection rate of 88.8%, while bilateral detection rate was 68.2%. According to published data, the reported overall detection rate varied from 86% to 97.4%; however, bilateral detection rates were usually lower, and varied from 52% to 96% [5,6,7,8,9,10,11,17,18,19]. These results came from the trials where both robotic and laparoscopic approaches have been used. In the studies where only laparoscopic surgery was performed, the overall detection rate varied from 95.5% to 97.4%, while bilateral detection rates were 76.4–88% [10,17,18,19]. However, compared to our practice, most of these studies used different methodology for ICG dye dilution and injection. We chose the methodology described by B. Geppert et al., [13] since they have demonstrated explicitly good bilateral SL detection rates (overall up to 100% and bilateral–98%), not only in one particular study, but in later trials as well [8,11]. Nonetheless, we could not achieve the reported rates.

Another important factor was the learning process for the new technique adopted. Our study was performed in a tertiary Oncogynaecological center with no previous experience on the SLB method in EC. Our results showed that the overall and bilateral detection rates improved over the period of the study, reaching up to 91.5% and 71.2%, respectively, in the last period of the study, and these results were comparable to those reported in the literature mentioned previously.

The distribution of mapped SLs was quite similar in both sides of the pelvis. The most common anatomical sites were the external iliac (45.8% on the right and 46.6% on the left) and obturator regions (20.9% and 25.6%, respectively). These results were consistent with those reported in the literature, where the rate of external iliac site SL was 32–59.3%, and obturator region was 18–25% [10,17,18,19].

Geppert B. and Persson J. with colleagues discussed two different lymphatic pathways of the uterine lymphatic system—the upper paracervical pathway, that drains to the obturator and the external iliac region, and the lower paracervical pathway that drains through the presacral region to the internal iliac site. The authors emphasized that both pathways should be visualized during SL mapping, while noting that lymph-node metastasis in uterine cancer could be detected in the presacral region as well [13]. In a prospective cohort study by Rossi et al. (FIRES trial), 17% of patients had positive SLs in the presacral and internal-iliac regions [5]. In our study, up to 19.0% of SL were identified in the internal iliac region. Out of eight patients with lymph-node metastasis, only one (12.5%) had metastasis in the internal iliac region in the ICG mapped SL; there was also one case where a successfully mapped SL in the common iliac region was proven to be metastatic.

The most common sites of metastatic lymph-nodes were the external iliac region and obturator fossa—35.7% in both places, respectively. These results were in accordance with previously reported metastatic lymph-node sites, where the rate of positive lymph-nodes in the external iliac region varies from 25% up to 41% and in the obturator fossa—25–60% [5,8,11].

We reported that the procedure had 75% sensitivity with a 97.2% negative predictive value. There were two cases where SL was negative for metastasis, but positive lymph-nodes were detected in full lymphadenectomy samples. In one of these cases, a negative SL was mapped in the common iliac region, while the metastatic lymph-node was found in the ipsilateral external iliac region. And in the other case—a SL mapped in the obturator site was negative, but metastasis was found in another non-mapped lymph-node of the same region.

We almost reached the reported sensitivity of previous studies, varying from 77.8% up to 98% [8,9]. The trend of FNR looks promising, as we managed to decrease it from 33.0% to 16.6% in the middle of our trial; however, it rose to 25.0% at the end. FNR highly depends on the number of patients with positive lymph-nodes. We can speculate that the better results could be achieved with higher number of cases with pathological lymph-nodes and additional experience on SL mapping in EC. The latter factor has been associated with successful bilateral mapping by some authors. The results of the study performed by Ianieri M. et al., showed that the surgeon was the only independent factor associated with the success of SLB [20]. As mentioned previously, the overall and bilateral detection rate in our study tended to increase with the experience as well and could be considered in agreement with this statement. However, the trial, performed by Sozzi G. and colleagues concluded that there are other factors supposedly associated with the failure of SL mapping as non-endometrioid tumour’s histology, lymphovascular space invasion, and bulky nodes [19]. We believe that this obstacle, together with the learning process of SLB procedure, should be further investigated to ensure a better performance.

The comprehensive histological evaluation of lymph-nodes has been demonstrated to be important. Some authors reported that after SL ultra-staging, the sensitivity of the procedure could increase up to 96.3–100% [6,7,11]. In our study, ultra-staging was not performed, and this may be considered as the major weakness of our trial.

In our trial only one patient (0.6%) had lymphatic complications after SLB + LND—vaginal lymphorrhea. No complications were noticed in the SLB-only group. The results of longitudinal prospective studies, regarding post-operative LND related complication in EC patients, showed that LND remained the biggest risk factor for lower limb lymphedema, a complication that had a negative impact on the patient’s physical well-being and body-image perception [3,4]. Geppert B. et al., emphasized that robotic SLB with ICG had a 14-fold decreased risk for this complication, compared to full LND, stating that not only low-risk patients would benefit from this procedure, but high-risk patients with contraindications for full LND as well [12].

Concerning the pre-operative low-risk patients’ group, our achieved bilateral SL mapping rate was 76.2% and the overall detection was up to 93.7%. After the final pathological evaluation, out of 63 patients, 6 (9.5%) were upstaged to intermediate and 8 (12.7%)—to high-risk disease groups. One patient had metastatic disease detected in a successfully mapped SL lymph-node. No lymphatic complications were documented in this patient group during the early post-operative period. While the SLB procedure was still debatable in this group, considering high success rates and low morbidity, we strongly believed that they would benefit from the SLB procedure.

## 5. Conclusions

Even in the center with no previous experience, sentinel lymph-node biopsy using ICG mapping is feasible. However, the favorable outcomes might be associated with the learning process of newly established method and future studies are needed to address this issue.

## Figures and Tables

**Figure 1 medicina-58-00712-f001:**
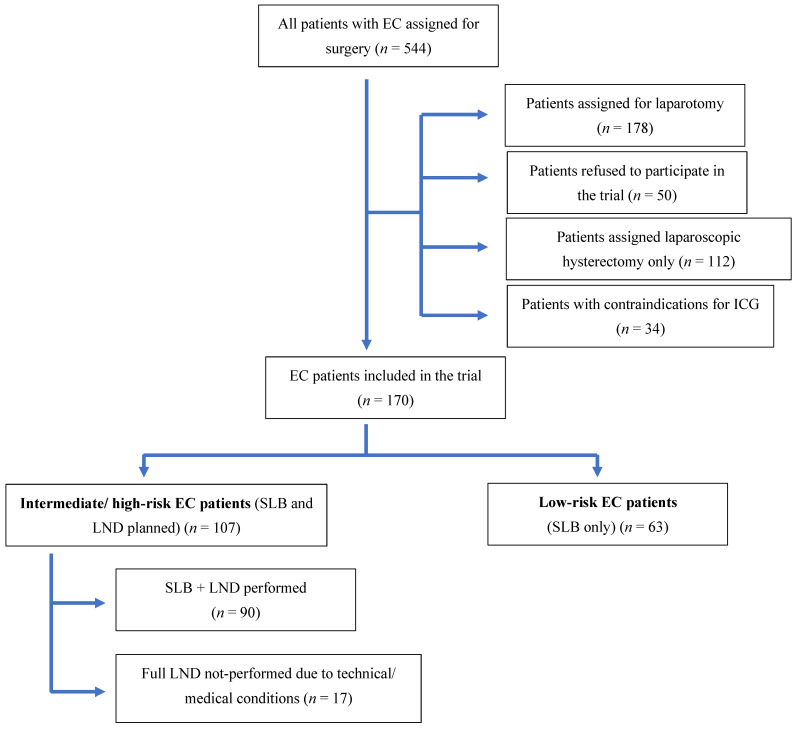
Flow-chart of patients’ selection. Abbreviations: EC—endometrial cancer, ICG—indocyanine green; SLB—sentinel lymph-node biopsy, LND—lymphonodectomy.

**Figure 2 medicina-58-00712-f002:**
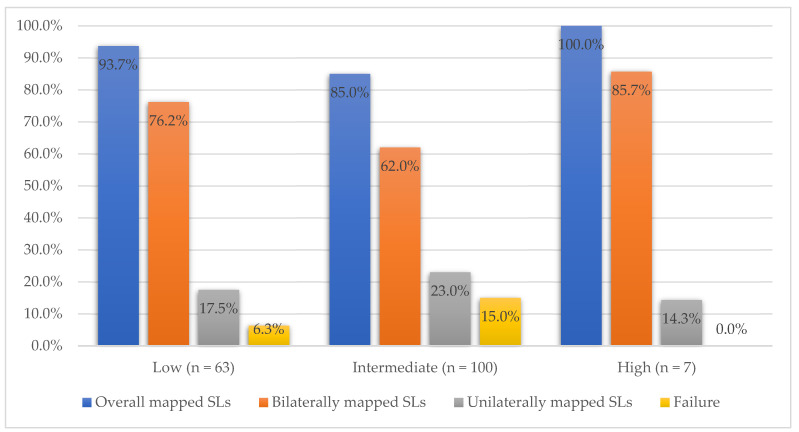
Sentinel lymph-nodes detection rates according to pre-operative risk.

**Figure 3 medicina-58-00712-f003:**
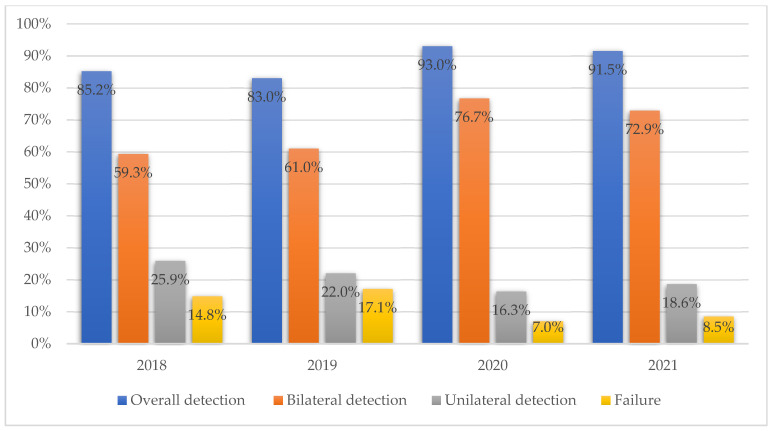
SL detection rates according to study period.

**Figure 4 medicina-58-00712-f004:**
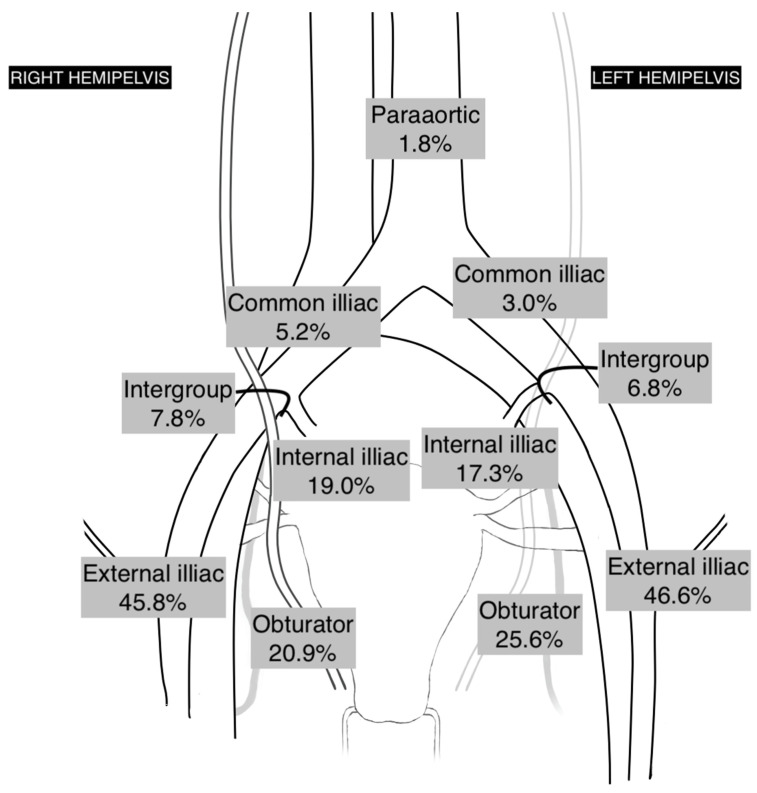
Anatomical sites of mapped SLs.

**Table 1 medicina-58-00712-t001:** General characteristics of the study population.

Parameter	Value
Age, years (Median, Interval)	63.5 (43.0–88.0)
Weight, kg (Median, Interval)	80.0 (46.0–169.0)
BMI (Median, Interval)	30.05 (19.23–48.05)
Morbid obesity, BMI > 40 (*n*, %)	15 (8.8%)
Metabolic syndrome (*n*, %)	30 (17.6%)
Extragenital disease (*n*, %):	
Hypertension	81 (47.6%)
Diabetes	3 (1.8%)
Ischaemic heart disease	11(6.5%)
Heart failure	4 (2.4%)
Hypertension and diabetes	14 (8.2%)
Other	9 (5.3%)
Pre-operative risk (*n*, %):	
Low	63 (37.1%)
Intermediate	100 (58.8%)
High	7 (4.1%)

**Table 2 medicina-58-00712-t002:** Parameters of surgical performance.

Parameter	SLB + LND (*n* = 90)	SLB Only (*n* = 80)
Surgery time, min (Median, Interval)	180 (90–455)	150 (105–300)
Blood loss, mL (Median, Interval)	50 (10–200)	50 (10–300)
Post-surgical complication— Clavien-Dindo classification (*n*, %):		
Grade I	1 (1.1%)	–
Grade II	3 (3.3%)	–
Grade III	2 (2.2%)	1 (1.3%)

**Table 3 medicina-58-00712-t003:** SL anatomical site depending on tumour histological type.

SL Anatomical Site	Endometrioid Type Tumours	Non-Endometrioid Type Tumours
Obturator region	21.8%	50.0%
External iliac region	45.2%	25.0%
Internal iliac region	19.5%	12.5%
Common iliac region	4.2%	12.5%
Iliac bifurcation	8.0%	–
Para-aortic region	1.1%	–

**Table 5 medicina-58-00712-t005:** Trend of FNR over the study period.

	Study Period			
	2018	2019	2020	2021
FNR (%)	33.0 %	33.0 %	16.6 %	25.0%

**Table 6 medicina-58-00712-t006:** Post-operative tumour evaluation.

Parameter	Total *n*, (%)
Tumour histological type:	
Endometrioid adenocarcinoma:	163 (95.9%)
G1	62 (36.5%)
G2	90 (52.9%)
G3	11 (6.5%)
Non-endometrioid type of tumour (total *n*, %):	7 (4.1%)
Myometrial invasion:	
Less than half of myometrium	96 (56.5%)
More than half of myometrium	74 (43.5%)
Lympho-vascular space invasion present	33 (19.4%)
Cervical stroma invasion	12 (7.1%)
Post-operative risk assessment:	
Low	88 (51.8%)
Intermediate	40 (23.5%)
High	42 (24.7%)

**Table 7 medicina-58-00712-t007:** Risk assessment before and after the surgery.

		Post-Operative risk		
		Low	Intermediate	High
Pre-operative risk	Low (total 63)	49 (77.8%)	6 (9.5%)	8 (12.7%)
	Intermediate (total 100)	39 (39.0%)	33 (33.0%)	28 (28.0%)
	High (total 7)	0	1 (14.3%)	6 (85.7%)

## Data Availability

Not applicable.

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
