# Peer review of "The Feasibility of Sentinel Lymph-Node, Mapped with Indocyanine Green, Biopsy in Endometrial Cancer Patients: A Prospective Study"

_medicina, 2022, doi:10.3390/medicina58060712_

Round 1
Reviewer 1 Report
Dear Authors,
This is a prospective study of 170 laparoscopically staged EC patients, to determine the feasibility of ICG mapped SLB in low, intermediate, and high-risk EC patients in a centre with no previous experience of this procedure.
This study is similar to numerous studies done so far.
Feasibility of a study means whether the study can be practically done and is it safe to perform the study. The conclusions quote” It allows to tailor the adjuvant treatment while avoiding complications related to systemic lymphadenectomy”. This cannot be concluded by the present study done
In the methods sections, the methods of sentinel node dissection should be briefly summarised. How many minutes after injection the nodes were dissected, whether you dissected both upper and lower paracervical pathway in all patients.
The false negative rate deserves a mention. The trend of FNR in the years of expertise also would be interesting for the reader in a first experience canter.
It is always interesting to know that after gaining experience, if you can share pearls of wisdom on how you could have reduced your FNR and why your mapping rates remained low.
The median number of removed lymph-nodes was 7.5 (interval 3 – 22). Its my assumption that this is full LN removal numbers. This appears less.
Rest, seems alright and this study will add to the similar studies , however, this doesn’t add any new perspective to our knowledge.
Regards
I am interested to know that how the factors of X1, X2, X3, X4 and X5 were chosen. You have mentioned that the post operative complications studied were taken from the previous literature however the factors and relationship to complications were discussed very late in the discussion.. The reader should know in the methods that why specifically these factors are studied.. The last few lines of section 3.1 looks incomplete or jumbled, may be because the units are not specified. Another thing is that how can this model help the surgeon practically. There is no cut off given for high or normal values of factors. I don't think that these statistical test can be applied in the clinic and this can give a comparative likelihood only instead of specific risk in terms of percentage, ratio etc .
In my opinion the manuscript should be improved to make it more interest generating and to help future studies on the risk assessment of complications.
Regards
Reviewer 2 Report
The prevalence of lymph-node is a common problem in endometrial cancer and other cancer types. Systemic pelvic lymphadenectomy (LND) is an important procedure normally used to evaluate the status of lymph-nodes in endometrial cancer (EC) patients, but it is commonly associated with post-surgical complications. Here, the authors have attempted to evaluate the status of lymph nodes in EC patients using Indocyanine green (ICG) in EC patients with the aim of reducing complications associated with LND in a center which has no prior experience of this procedure. The authors were able to obtain an overall SL detection rate of 88.8%. They also achieved bilateral mapping in 68.2%. Most importantly, the overall detection rate was 100% in high-risk patients, 93.7% in low-risk patients and 85% in intermediate group. The study is interesting; however, the authors can strengthen the study by addressing the following points –
Minor Points –
- The authors should add a paragraph in the discussion about the possibility of the ICG procedure in patients with other cancers like breast cancer which also involves lymph node metastasis.
- PTEN mutations are most common in endometrial cancer. Were patients in the study evaluated for PTEN, BRCA1/2 mutations? If yes, then is there a correlation between PTEN, BRCA1/2 mutation and low risk, intermediate vs high risk?
- In Figure 2, there is no label on the y-axis. The authors must label the y-axis. The authors must also include unilateral detection in this graph.
- In addition to Figure 2, it will be more informative if the authors plot a graph of SL detection rates in low, intermediate, and high pre-operative risk group (Plot a graph of numbers in table 3).
- The authors have shown the anatomical sites of mapped SLs in Figure 3. The authors should comment if there is a correlation/pattern between tumor histological type and anatomical sites of mapped SLs? Adding this information will be more informative.
